# Research Trends and Hotspots of Retinal Optical Coherence Tomography: A 31-Year Bibliometric Analysis

**DOI:** 10.3390/jcm11195604

**Published:** 2022-09-23

**Authors:** Aidi Lin, Xiaoting Mai, Tian Lin, Zehua Jiang, Zhenmao Wang, Lijia Chen, Haoyu Chen

**Affiliations:** 1Joint Shantou International Eye Center, Shantou University and the Chinese University of Hong Kong, Shantou 515041, China; 2Department of Ophthalmology & Visual Sciences, The Chinese University of Hong Kong, Hong Kong, China

**Keywords:** retinal optical coherence tomography, trends, hotspots, bibliometric analysis

## Abstract

The emergence of optical coherence tomography (OCT) over the past three decades has sparked great interest in retinal research. However, a comprehensive analysis of the trends and hotspots in retinal OCT research is currently lacking. We searched the publications on retinal OCT in the Web of Science database from 1991 to 2021 and performed the co-occurrence keyword analysis and co-cited reference network using bibliometric tools. A total of 25,175 publications were included. There has been a progressive increase in the number of publications. The keyword co-occurrence network revealed five clusters of hotspots: (1) thickness measurements; (2) therapies for macular degeneration and macular edema; (3) degenerative retinal diseases; (4) OCT angiography (OCTA); and (5) vitrectomy for macular hole and epiretinal membrane. The co-citation analysis displayed 26 highly credible clusters (S = 0.9387) with a well-structured network (Q = 0.879). The major trends of research were: (1) thickness measurements; (2) therapies for macular degeneration and macular edema; and (3) OCTA. Recent emerging frontiers showed a growing interest in OCTA, vessel density, choriocapillaris, central serous chorioretinopathy, Alzheimer’s disease, and deep learning. This review summarized 31 years of retinal OCT research, shedding light on the hotspots, main themes, and emerging frontiers to assist in future research.

## 1. Introduction

Optical coherence tomography (OCT) is a widely used technology for high-resolution and cross-sectional imaging of tissues by measuring backscattered light [1]. In 1991, the first report on retinal imaging by OCT in vitro was published [2]. Since then, OCT has undergone substantial progress. As the first generation of OCT systems, time-domain OCT has appeared as a diagnostic tool and revealed the pathogenesis of macular diseases [1]. Application of OCT for detecting and monitoring glaucoma was reported from the beginning of the retinal nerve fiber layer (RNFL) thickness measurements [3]. The advent of spectral-domain OCT (SD-OCT) has enabled the segmentation of selected layers and visualization of anatomic landmarks [4]. Moreover, various OCT signs may serve as indicators of disease severity, treatment response, and prognostic prediction. For example, the absence of intraretinal cysts has been proven to predict spontaneous closure of traumatic macular hole (MH) [5]. Recent advancements in OCT techniques, including enhanced depth imaging OCT and swept-source OCT, have facilitated in-depth analysis of the choroid. More recently, OCT angiography (OCTA) has been developed to generate high-resolution retinal microvasculature images [6]. The quantitative analysis of OCTA images enables the evaluation of vascular abnormalities in retinal vascular diseases [7].

The rapid development of OCT has sparked great interest in retinal research over the years. Thousands of publications have reported the advanced OCT technology and clinical applications, and systematic reviews and meta-analyses have focused on specific questions of retinal OCT research [8,9]. With the significant growth in the production of research literature, novel approaches are required to review and analyze the trends within the domain of retinal OCT knowledge [10].

Bibliometric analysis is a quantitative research method that extracts measurable data from publications through mathematical and statistical methods. It allows researchers to visualize how pieces of evidence are interconnected, the knowledge structure and research trends within a body of literature. The statistical indicator can also objectively measure the scholarly impact of publications produced in this field [11]. Nowadays, bibliometric analyses have become increasingly popular in ophthalmology [12,13]. Though multiple reviews with different emphases on retinal OCT have been published [14,15], a comprehensive analysis of trends and hotspots of retinal OCT is still lacking. Therefore, we performed a bibliometric analysis of retinal OCT literature from 1991 to 2021 to elucidate the evolution of trends, hotspots, and emerging frontiers within this rapidly expanding knowledge domain.

## 2. Materials and Methods

### 2.1. Search Strategy and Data Rretrieval

Our retrieval used the Science Citation Index Expanded in the Core Collection of Web of Science (WOS) on 27 February 2022. We performed a Boolean search on two lists with the operator “and”. The first list included the following search structure: TS = (“optical coherence tomography” OR “OCT angiography”). The second list of search structures was as below: TS = (macula* OR retina* OR retino* OR vitreal OR vitreo* OR choroid* OR “posterior segment” OR fundus OR “posterior pole”). We included the articles and reviews in all languages with the timespan from 1991 to 2021.

The retrieved articles were exported as “full records and cited references”. Moreover, we obtained the impact factor and journal quartile according to Journal Citation Reports’ 2020 standards.

### 2.2. Data Analysis

Different models were adopted to fit the time trend curve of the global annual number of publications, including the linear, logarithmic, and polynomial functions. We also used the WOS Analysis tool, VOSviewer (1.6.18, Centre for Science and Technology Studies, Leiden University, The Netherlands), and CiteSpace (5.8.R3, creator Chaomei Chen) to visualize the data.

#### 2.2.1. WOS Analysis Tool

The most productive countries/regions, institutions, authors, journals, and the H-index of country/region were presented as tables or graphs in this tool.

#### 2.2.2. VOSviewer

VOSviewer is a computer program for constructing and viewing maps using bibliometric data [16]. Based on the similarity of research topics, co-occurrence analysis can extract high-frequency keywords and group them as clusters for covering the hotspots. In the network visualization, each keyword is labeled as a node, and the weight of the occurrence determines its size. The links refer to the co-occurrence relationship between keywords.

We also used the VOSviewer to visualize the collaboration maps of the countries/regions, institutions, and authors. The size of an item is determined by the weight, while the color of an item is decided by the cluster to which the item belongs. The total link strength can reflect the cooperation intensity. A higher value indicates a more vigorous intensity of the cooperation relationship.

#### 2.2.3. CiteSpace

According to research fronts and intellectual bases, CiteSpace is a java software designed for the trends and transient patterns in scientific publications [10]. Co-citation represents the frequency of two references cited by another article; the more co-citations they receive, the more likely they are semantically associated. Therefore, a co-citation network can reveal the evolution of the research areas derived from the original publications.

We automatically labeled co-citation clusters by adopting the log-likelihood ratio algorithm. Citespace produces structural metrics, including centrality, modularity (or Q score), and silhouette score (or S score). Centrality measures the number of times a node lies on the shortest path connecting with other nodes. Nodes with high centrality values are key centers that link with different clusters. The Q score from 0 to +1 measures the extent to which clusters or modules can form a co-cited reference network. The cluster structure with a Q score exceeding 0.3 is significant, and higher values indicate a better-organized structure. The S score validates the consistency within clusters of data ranging from −1 to +1. Interpretations for the S score are: >0.3, homogenous; >0.5, reasonable; >0.7, highly credible. Burst detection can indicate abrupt surges in interest towards a particular node [17]. Besides, the timeline view of co-citation analysis can present the evolution of the research topics. Every horizontal row represents a cluster, and each node of a study is marked as a “tree ring” on the line.

## 3. Results

### 3.1. Publication Output and Growth Trend

Between 1991 to 2021, 25,175 publications on retinal OCT were searched, including 23,654 articles and 1521 reviews. As shown in Figure 1, the global annual number of publications has an increasingly growing trend. The polynomial function was the best among different prediction models, with the highest R^2^ for the polynomial function being 0.9956, linear function 0.8499, and logarithmic functions 0.5261. According to the prediction function (y = 4.9473x^2^ − 62.849x + 155.4), about 3210 papers are projected to be published in 2022.

### 3.2. Co-Occurrence Analysis of the Top 100 Keywords

Figure 2 presents a co-occurrence network of the top 100 keywords. Notably, the keywords OCT (11,305), thickness (2313), macular degeneration (1691), degeneration (1055), and OCTA (1175) were located at the center of the map. Based on the similarity of the research topics, the keywords were grouped into 5 clusters with different colors in the network. Cluster 1 in red represented the thickness measurements (Appendix A), cluster 2 in green represented the therapies for macular degeneration and macular edema (Appendix A), cluster 3 in blue represented the degenerative retinal diseases (Appendix A), cluster 4 in yellow represented the OCTA technique (Appendix A), and cluster 5 in purple represented the vitrectomy for MH and epiretinal membrane (ERM) (Appendix A).

To explore the evolution trends over time, the keyword co-occurrence analysis was performed for publications in each decade separately (Appendix A). In the first decade (Appendix A: 1991–2001), the clusters of thickness measurements (cluster 1 in red), vitrectomy for MH and ERM (cluster 2 in green), and degenerative retinal diseases (cluster 8 in brown) appeared, which kept in growing in the next two decades. In the second decade (Appendix A: 2002–2011), the cluster of therapies for macular degeneration and macular edema (cluster 1 in red) developed. Subsequently, the cluster of the OCTA technique (cluster 3 in blue) emerged in the third decade (Appendix A: 2012–2021).

### 3.3. Co-Citation Reference Analysis

#### 3.3.1. Clusters of Research

As presented in Figure 3A, the co-citation reference map showed a well-structured network and highly credible clusters, respectively (Q = 0.879; S = 0.9387). It consisted of 26 different clusters with the indications of the label, size, silhouette score, and the mean year of publication of the cluster members (Appendix A). The timeline view (Figure 3B) demonstrated the evolution of research topics over time.

Three different major trends composed of multiple clusters separately were identified. The first trend concerned the thickness measurements (corresponding to cluster 1 in Figure 2). It started with the research on retinal thickness (‘cluster #7’: 130; S = 0.866; 2003), which further evolved into glaucoma (‘cluster #4’: 173; S = 0.932; 2011), and choroidal thickness (‘cluster #6’: 135; S = 0.971; 2011).

The second major trend focused on the therapies for macular degeneration and macular edema (corresponding to cluster 2 in Figure 2). It began with research on photodynamic therapy (‘cluster #21’: 50; S = 0.976; 2001), then developed into research on triamcinolone acetonide (‘cluster #19’: 53; S = 0.976; 2003). Later, the research on intravitreal anti-vascular endothelial growth factor (anti-VEGF) agents appeared, including the bevacizumab (‘cluster #3’: 183; S = 0.888; 2006) and aflibercept (‘cluster #20’: 52; S = 0.952; 2012). More recently, this trend evolved into the research on diabetic macular edema (DME) (‘cluster #23’: 44; S = 0.995; 2014).

The third major trend concentrated on the OCTA (corresponding to cluster 4 in Figure 2). As the largest cluster in the past 31 years, the research on OCTA (‘cluster #0’: 297; S = 0.889; 2015) developed with a paper by Jia et al., published in 2012 (centrality: 0.12). Currently, this cluster further branched into the research on vessel density (‘cluster #16’: 75; S = 0.976; 2016) and choriocapillaris (‘cluster #13’: 94; S = 0.956; 2017).

In addition, the emerging frontiers were OCTA (‘cluster #0’: 297; S = 0.889; 2015), vessel density (‘cluster #16’: 75; S = 0.976; 2016), choriocapillaris (‘cluster #13’: 94; S = 0.956; 2017), central serous chorioretinopathy (‘cluster #15’: 84; S = 0.957; 2015), Alzheimer’s disease (‘cluster #14’: 93; S = 0.979; 2015), and deep learning (‘cluster #9’: 115; S = 0.956; 2015).

#### 3.3.2. Most Co-Cited Papers and Burst Detection

Among the top 10 most co-cited references (Table 1), seven of them were related to the cluster of OCTA [6,7,15,18,19,20,21]. The first three most co-cited papers were published by Spaide et al. The comparative study of fluorescein angiography and OCTA in the retinal vasculature imaging published in JAMA Ophthalmology was the most cited article with 570 citations [18]. The second most cited paper, published in Retina-The Journal of Retinal and Vitreous Diseases, established a framework for the image artifacts of OCTA with 417 citations [15]. In 2018, a review on OCTA published in Progress in Retinal and Eye Research ranked third among the most co-cited references with 356 citations [6].

The top 25 citation bursts detected from 1991 to 2021 are illustrated in Figure 4. The strongest citation burst (strength = 164.78) was observed for the paper published by Jia et al., that lasted from 2015 to 2017 [19]. Eight articles showed a currently active citation burst, all of which focused on the topic of OCTA: a comparative study of retinal vascular layer imaging between OCTA and fluorescein angiography [18]; a guideline to recognize OCTA image artifacts [15]; a quantitative OCTA analysis of vascular abnormalities [7]; a clinical evaluation of ocular perfusion in glaucoma using OCTA [24]; a detailed presentation of retinal vascular anatomy by OCTA [21]; and three reviews of OCTA published in 2015 [14], 2017 [25], and 2018 [6], respectively.

### 3.4. Contribution of Countries/Regions, Institutions, and Authors

A total of 120 countries/regions have been involved in this field. Table 2 presents the top 10 productive countries/regions. The United States had the most publications (7835 papers), followed by China (2705 papers) and Japan (2269 papers). The total citations (295,069 times), average article citations (37.66 times), and H-index (198) in the United States far exceeded other countries/regions in the world. Figure 5 shows the analysis of the top 10 productive countries on an annual basis from 1991 to 2021. The United States ranked first each year, while China obtained the most rapid increase in recent years.

We found 9030 institutions and more than 49,000 authors. As presented in Table 3, the University of California System from the United States was the most productive (1272 papers), followed by the University of London (1011 papers), and the University College London (946 papers) from England. Among the top 10 contributing scholars (Table 4), Bandello F ranked first with 299 papers, followed by Querques G (249 papers), and Schmidt-erfurth U (239 papers).

As shown in the collaboration network, the most collaborative country/region, institution, and author were the United States (Figure 6A and Table 2; total link strength = 5300), the University of California Los Angeles (Figure 6B and Table 3; total link strength = 1507), and Bandello F (Figure 6C and Table 4; total link strength = 1202), respectively.

### 3.5. Contribution of Journals

Table 5 shows the top 10 productive and co-cited journals. Investigative Ophthalmology and Visual Science obtained the most publications (2026), followed by Retina-The Journal of Retinal and Vitreous Diseases (1994) and American Journal of Ophthalmology (1260). The top 10 contributing journals were classified as Q1 or Q2, except for European Journal of Ophthalmology (Q3). Of the top co-cited journals, Investigative Ophthalmology and Visual Science also ranked first with 90,587 citations, followed by Ophthalmology (83,931 citations) and American Journal of Ophthalmology (67,517 citations). All the top 10 co-cited journals were classified as Q1 or Q2, with the impact factor ranging from 3.117 to 21.198.

## 4. Discussion

We conducted a bibliometric analysis of publications within the past 31 years on retinal OCT to gauge the progress made so far and to detect emerging trends.

### 4.1. Global Output on Retinal OCT Research

The number of publications on retinal OCT research has risen continually since 1991, and nearly half of the articles have been produced over the last five years. As of 2021, the number of publications exceeded 3000 per year. It is likely to keep rising according to the polynomial prediction function. More than 49,000 authors from 9030 institutions in 120 countries have published articles in this field. Among all the countries, the United States plays a leading role in the quantity and quality of the publications, showing the most publications, total citations, and H-index. Recently, China has presented the most rapid increase in annual publications (Figure 5), accounting for a gradual decrease in the gap between the United States and other countries.

### 4.2. Trends and Hotspots in Retinal OCT Research

The keyword co-occurrence analysis (Figure 2) has favored classifying the knowledge structure and hotspots.

#### 4.2.1. Cluster 1 (Figure 2, Red Cluster): Thickness Measurements by OCT

This cluster describes the thickness measurements as the first major trend in retinal OCT research. Retinal thickness has been considered a marker for disease severity [1], defined as the distance between the internal limiting membrane (ILM) and the retinal pigment epithelium (RPE). This metric has been commonly used to evaluate the retinal morphological changes after the treatments. The reproducibility of thickness measurements has been well investigated since it is an essential quality to determine the utility of a device for clinical use [26].

The success of OCT in detecting glaucomatous structural damage began with the measurements of RNFL thickness by taking glaucoma from a primarily subjectively evaluated disease to an objectively assessed disease [3]. In 2005, the first report to measure longitudinal changes of RNFL thickness in glaucoma was published using time-domain OCT [37]. Later, the emergence of SD-OCT can achieve higher reproducibility of RNFL measurements and improve the ability to detect glaucoma progression [38]. Besides, other retinal layers, such as the ganglion cell-inner plexiform layer, have been measured for early glaucoma detection.

In 2008, Spaide et al., proposed the enhanced depth imaging OCT to obtain detailed choroidal images and measure the choroidal thickness, defined as the vertical distance from the posterior edge of the RPE to the choroid/sclera junction [30]. Choroidal thickness has become a quantitative biomarker for choroidal tissues suggestive of different pathogenesis in various diseases. For example, choroidal thickening has been observed in polypoidal choroidal vasculopathy, in contrast with choroidal thinning in exudative age-related macular degeneration (AMD) [39].

#### 4.2.2. Cluster 2 (Figure 2, Green Cluster): Therapies for the Treatments of Macular Degeneration and Macular Edema

This cluster displays the management of macular degeneration and macular edema, emerging as the second major trend in retinal OCT. Macular degeneration, also denominated AMD, is a leading cause of blindness among the aging population in developed countries. The advances in chemistry and pharmacology have allowed for the effective treatments of the neovascular AMD, characterized by the formation of choroidal neovascularization. Verteporfin was the first agent for photodynamic therapy in AMD with predominantly classic subfoveal choroidal neovascularization [40]. In 2006, ranibizumab was approved and proved superior to verteporfin with low rates of severe ocular adverse events [23,27]. After that, other anti-VEGF agents, including bevacizumab and aflibercept, have shown similar treatment efficacy to ranibizumab [34,41]. With the anti-permeability effects, intravitreal anti-VEGF has become an effective treatment for neovascular AMD, and OCT has been extensively performed to detect retinal changes in therapeutic follow-up. Moreover, an OCT-guided, variable-dosing regimen with the intravitreal injection can be provided if retreatment with anti-VEGF is necessary [29].

Macular edema manifests as abnormal macular swelling and thickening associated with the accumulation of intra- or subretinal fluid [42]. It can occur in various pathologic conditions, including diabetic retinopathy, retinal vein occlusion, uveitis, and postsurgical inflammation. As a multifactorial pathologic example, DME is primarily due to the increased retinal capillary permeability in diabetic patients. Triamcinolone acetonide has been used as a corticosteroid agent for treating DME patients unresponsive to laser photocoagulation [43]. Later, the intravitreal anti-VEGF injection provided superior visual acuity gain over standard laser in DME [44]. However, the relative effects are dependent on baseline vision. In eyes with retinal vein occlusion, intravitreal anti-VEGF therapy has become the current standard of care in macular edema though photocoagulation, and corticosteroid therapies are reasonable in certain circumstances [45]. In pseudophakic macular edema, topical steroidal or nonsteroidal anti-inflammatory drugs, either separately or combined, have been demonstrated to be effective [46]. Remarkably, various OCT biomarkers have been extensively used to evaluate the severity of macular edema and treatment responses [47,48].

#### 4.2.3. Cluster 3 (Figure 2, Blue Cluster): Degenerative Retinal Diseases

Degenerative retinal diseases are heterogeneous and multi-etiological groups of disorders that will result in irreversible visual damage and compromised life quality [49]. The depth-resolved OCT allows us to identify the tissue loss layer by layer because the changes may vary among layers in these atrophic diseases.

Retinitis pigmentosa (RP) is one of the most common degenerative retinal diseases characterized by the degeneration of photoreceptor cells and RPE. The progressive loss of outer retinal layers has been demonstrated on OCT. At the early stage of RP, the optical intensity of the ellipsoid zone has proved to be an indicator of retinal degeneration [50]. As RP progresses, the thinning or loss of the outer segments may happen [51].

Geographic atrophy (GA) is an advanced form of AMD with the degeneration of photoreceptors and RPE. SD-OCT has become the most recent reference standard for GA assessment among the existing imaging modalities. An OCT-based classification system has been proposed to define atrophy. It can help recognize the biomarkers at different stages of atrophy, including incomplete and complete RPE and outer retinal atrophy [52]. To optimize the diagnosis and prognosis of AMD patients, automated segmentation and quantification of GA from OCT have been well investigated [53].

#### 4.2.4. Cluster 4 (Figure 2, Yellow Cluster): OCTA Technique

In 2012, a novel OCTA technique, namely the split-spectrum amplitude-decorrelation angiography, was developed with an improved signal-to-noise ratio of flow detection than other amplitude-decorrelation algorithms [19]. Though fluorescein angiography has been traditionally used for retinal vasculature evaluation, the imaging of OCTA shows the advantages of capturing all retinal vascular layers without dye injection [18]. Recently, OCTA has become a noninvasive and convenient technique for detailed imaging and quantitative evaluation of vascular abnormalities [7,20].

Vessel density was calculated as the percentage area occupied by blood vessels measured by OCTA. It has been found that vessel density was associated with the severity of visual field damage in glaucoma. This association was generally stronger than standard structural measures such as RNFL [54]. The quantification of vessel density has facilitated our understanding of the vasculature involved in the pathophysiology and improved the ability of disease monitoring [24].

Given that previous imaging techniques have limited the choriocapillaris imaging, the advent of OCTA is vital to present the choriocapillaris as a granular appearance [15]. The choriocapillaris enface images have demonstrated what appear to be areas of missing flow signal, known as signal voids or flow voids. Interestingly, the choriocapillaris signal voids have shown to follow a power law distribution, the alterations of which offer diagnostic possibilities and impact theories of disease pathogenesis [55]. Besides, choriocapillaris signal voids have appeared to be a valuable parameter for evaluating eyes with AMD. With deeper penetration than spectral-domain OCTA, swept-source OCTA has achieved reproducible imaging of the choriocapillaris and associated signal voids in eyes with drusen [56]. However, the choriocapillaris flow speeds or capillary leakage are still not provided by the current OCTA techniques, which may be promising to reveal the further pathogenesis in diseases affected by the choroid [57].

#### 4.2.5. Cluster 5 (Figure 2, Purple Cluster): Vitrectomy for MH and ERM

Pars plana vitrectomy is the primary treatment option for patients with MH. Recently, vitrectomy with ILM peeling has been recommended due to the improved visual and anatomic success compared with no ILM peeling [58]. For symptomatic ERM, vitrectomy with membrane peeling remains the mainstay of treatment, and sometimes additional ILM peeling is performed to reduce recurrence [59].

OCT imaging is essential in the preoperative and postoperative management of MH and ERM. Preoperatively, OCT has been utilized to identify the vitreomacular interface disorders, including MH and ERM. Moreover, multiple OCT parameters have prognostic value in the anatomic and visual outcomes. In eyes with MH, preoperative hole diameter, such as the base and minimum diameters determined by OCT, can predict the postoperative success rate of surgery [60]. For ERM, OCT biomarkers that were suggestive of a worse prognosis included the presence of cystoid macular edema, ectopic inner foveal layers, and cone outer segment termination defects [59]. Postoperatively, the anatomic outcomes can be evaluated using OCT, including the hole closure, membrane removal, traction relief, and retinal reattachment. Furthermore, the presence of subretinal fluid postoperatively can be detected by OCT, which may probably account for poor vision despite successful surgery.

### 4.3. Emerging Frontiers in Retinal OCT Research

In the past 31 years, the emerging frontiers of retinal OCT are OCTA, vessel density, choriocapillaris, central serous chorioretinopathy, Alzheimer’s disease, and deep learning. The emerging trend of the OCTA technique has developed the subtopics of vessel density and choriocapillaris that we have discussed before. Central serous chorioretinopathy is considered one of the pachychoroid spectrum disorders, characterized by the increased choroidal thickness and dilated outer choroidal vessels on OCT [57]. The emerging research on Alzheimer’s disease has benefited from the quantitative OCT/OCTA analyses that have extended into neurodegenerative disorders. Evidence has shown the retinal thickness and microvascular abnormalities associated with Alzheimer’s disease [9]. Nowadays, OCT/OCTA-based deep learning algorithms have been applied in the classification tasks for various diseases and segmentation tasks, including the delineation of macular edema [61]. The emerging frontiers of Alzheimer’s disease and deep learning suggest a multidisciplinary trend that may engage in retinal OCT research.

### 4.4. Limitations

Bibliometric analysis is relatively comprehensive and objective for exploring the scientific activities in a particular knowledge domain, but this study has some limitations. Firstly, we only extracted articles and reviews from 1991 to 2021; thus, those crucial articles with other document types or published in 2022 may be neglected. Secondly, we did not evaluate the quality of publications, then the articles with high and low qualities were given the same weight. Thirdly, the information from the WOS database was downloaded as “full records and cited references,” which may omit some valuable details or opinions. Although the WOS database is the most commonly used and recommended database for bibliometric analysis [12], some vital publications may not be included in this database. Therefore, other databases such as PubMed or Scopus should be adopted for further investigations.

## 5. Conclusions

In conclusion, this study comprehensively summarizes and visualizes retinal OCT research from 1991 to 2021, including publication outputs, hotspots, major trends, the latest topics, and global contribution networks. These findings, from a bibliometric perspective, will assist in identifying the evolution and emerging trends for future research.

## Figures and Tables

**Figure 1 jcm-11-05604-f001:**
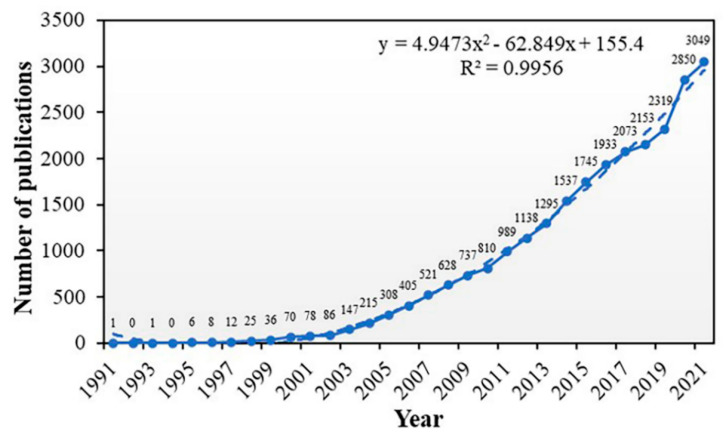
Number of publications worldwide from 1991 to 2021. Fitting formula (dotted line): y = 4.9473x^2^ − 62.849x + 155.4 (R^2^ = 0.9956).

**Figure 2 jcm-11-05604-f002:**
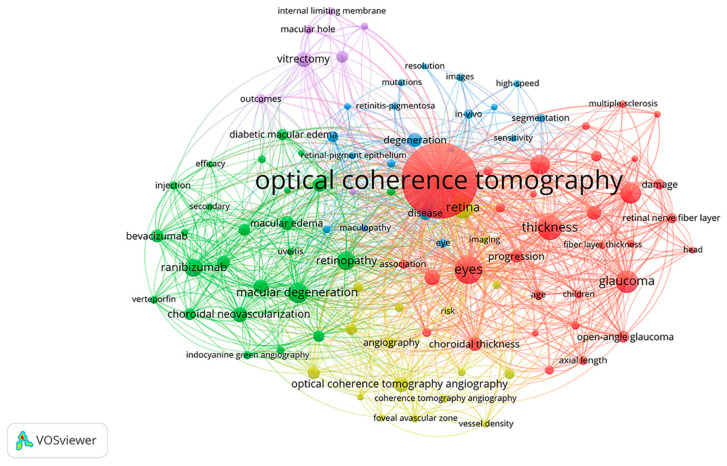
Co-occurrence analysis of the top 100 keywords.

**Figure 3 jcm-11-05604-f003:**
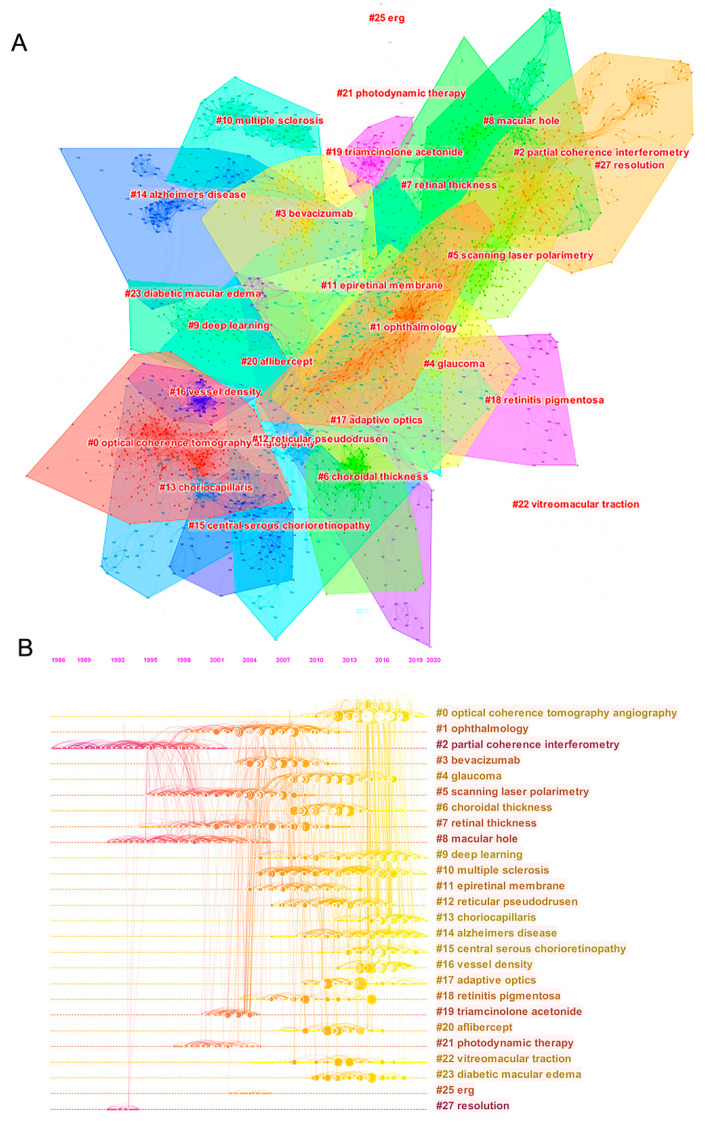
Co-citation reference and corresponding clustering analysis. (**A**) The network map of co-citation clusters; (**B**) The timeline view of co-citation clusters.

**Figure 4 jcm-11-05604-f004:**
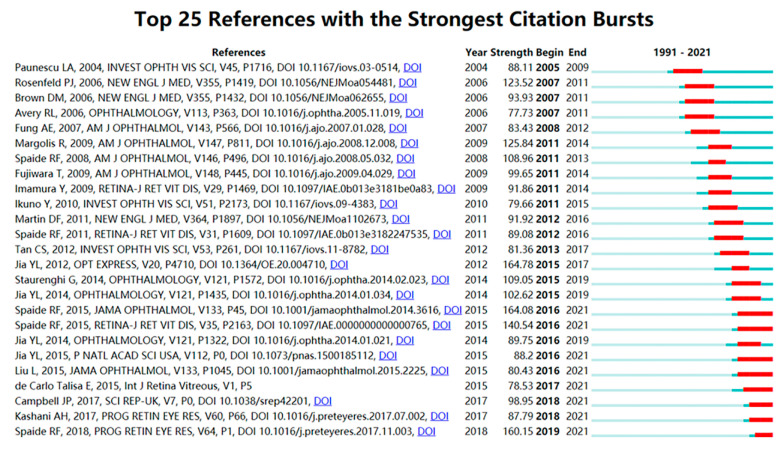
Top 25 references with the strongest citation bursts [4,6,7,14,15,18,19,20,21,22,23,24,25,26,27,28,29,30,31,32,33,34,35,36].

**Figure 5 jcm-11-05604-f005:**
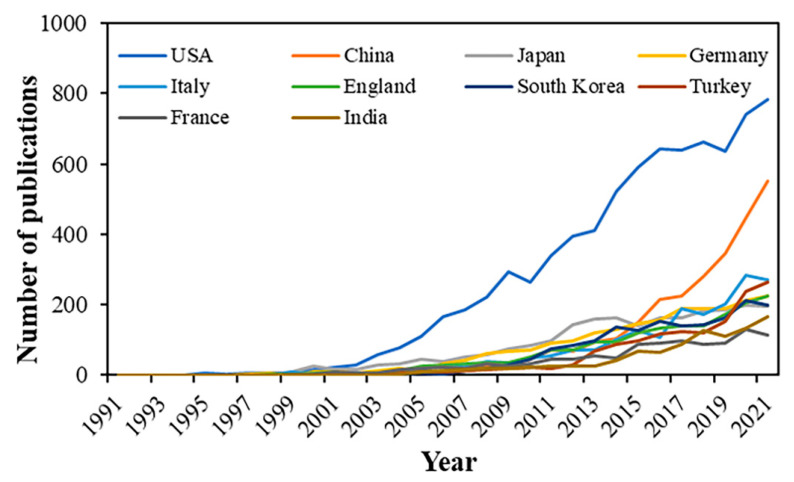
Number of publications of the top 10 productive countries/regions.

**Figure 6 jcm-11-05604-f006:**
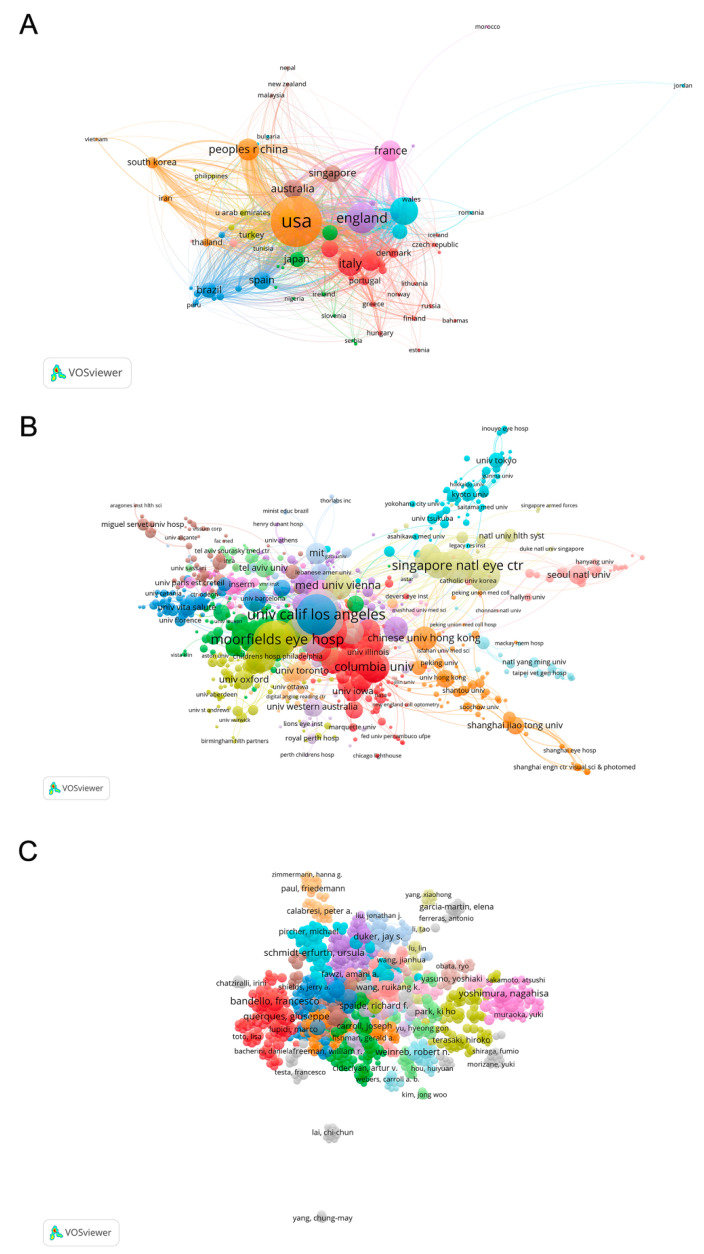
Analysis of Co-authorship relationship. (**A**) Collaboration network of countries/regions; (**B**) Collaboration network of institutions; (**C**) Collaboration network of authors.

**Table 1 jcm-11-05604-t001:** The top 10 most co-cited references.

Rank	First Author	Year	Source	Title	Doi	Citations	Cluster
1 [18]	Spaide RF	2015	JAMA Ophthalmol.	Retinal vascular layers imaged by fluorescein angiography and optical coherence tomography angiography	10.1001/jamaophthalmol.2014.3616	570	0
2 [15]	Spaide RF	2015	Retina-J. Ret. Vit. Dis.	Image artifacts in optical coherence tomography angiography	10.1097/IAE.0000000000000765	417	0
3 [6]	Spaide RF	2018	Prog. Retin. Eye Res.	Optical coherence tomography angiography	10.1016/j.preteyeres.2017.11.003	356	0
4 [19]	Jia YL	2012	Opt. Epress	Split-spectrum amplitude-decorrelation angiography with optical coherence tomography	10.1364/OE.20.004710	340	0
5 [4]	Staurenghi G	2014	Ophthalmology	Proposed lexicon for anatomic landmarks in normal posterior segment spectral-domain optical coherence tomography: the IN•OCT consensus	10.1016/j.ophtha.2014.02.023	308	17
6 [20]	Jia YL	2014	Ophthalmology	Quantitative optical coherence tomography angiography of choroidal neovascularization in age-related macular degeneration	10.1016/j.ophtha.2014.01.034	290	0
7 [22]	Margolis R	2009	Am. J. Ophthalmo.l	A pilot study of enhanced depth imaging optical coherence tomography of the choroid in normal eyes	10.1016/j.ajo.2008.12.008	274	6
8 [7]	Jia YL	2015	Proc. Natl. Acad. Sci. USA	Quantitative optical coherence tomography angiography of vascular abnormalities in the living human eye	10.1073/pnas.1500185112	272	0
9 [21]	Campbell JP	2017	Sci. Rep.	Detailed Vascular Anatomy of the Human Retina by Projection-Resolved Optical Coherence Tomography Angiography	10.1038/srep42201	264	0
10 [23]	Rosenfeld PJ	2006	N. Engl. J. Med.	Ranibizumab for neovascular age-related macular degeneration	10.1056/NEJMoa054481	252	3

**Table 2 jcm-11-05604-t002:** The top 10 most productive and collaborative countries/regions.

Rank	Country/Region	Record	Citations	Average Article Citations	H-Index	Rank	Co-Authorship Country/Region	Total Link Strength
1	USA	7835	295,069	37.66	198	1	USA	5300
2	China	2705	41,667	15.4	86	2	England	2242
3	Japan	2269	57,132	25.18	100	3	Germany	1937
4	Germany	2106	56,136	26.66	103	4	Italy	1587
5	Italy	1879	37,785	20.11	80	5	China	1335
6	England	1715	47,537	27.72	97	6	France	1204
7	South Korea	1644	29,381	17.87	67	7	Switzerland	1139
8	Turkey	1413	12,076	8.55	41	8	Australia	1006
9	France	1083	27,380	25.28	78	9	India	936
10	India	975	13,261	13.6	54	10	Spain	931

**Table 3 jcm-11-05604-t003:** The top 5 most productive and collaborative institutions.

Rank	Institutions	Record	Countries	Rank	Co-Authorship Institution	Total Link Strength	Countries
1	University of California System	1272	USA	1	University of California Los Angeles	1507	USA
2	University of London	1011	England	2	Moorfields Eye Hospital	1339	England
3	University College London	946	England	3	University College London	1317	England
4	Moorfields Eye Hospital NHS Foundation Trust	777	England	4	Vitreous Retina Macula Consultant of New York	1206	USA
5	Medical University of Vienna	564	Austria	5	New York University	1181	USA

**Table 4 jcm-11-05604-t004:** The top 10 most productive and collaborative authors.

Rank	Author	Records	Rank	Co-Authorship Author	Total Link Strength
1	Bandello F	299	1	Bandello F	1202
2	Querques G	249	2	Yoshimura N	1174
3	Schmidt-erfurth U	239	3	Querques G	1088
4	Sadda SR	235	4	Tsujikawa A	786
5	Yoshimura N	227	5	Duker JS	760
6	Freund KB	215	6	Fujimoto JG	703
7	Duker JS	207	7	Weinreb RN	696
8	Holz FG	193	8	Holz FG	636
9	Weinreb RN	188	9	Huang D	636
10	Fujimoto JG	185	10	Schmidt-erfurth U	628

**Table 5 jcm-11-05604-t005:** The top 10 productive and co-cited journals.

Rank	Journal	Record	Impact Factor	Journal Quartile	Rank	Co-Cited Journal	Cited Time	Impact Factor	Journal Quartile
1	Invest. Ophth. Vis. Sci.	2026	4.799	Q1	1	Invest. Ophth. Vis. Sci.	90,587	4.799	Q1
2	Retina-J. Ret. Vit. Dis.	1994	4.256	Q1	2	Ophthalmology	83,931	12.079	Q1
3	Am. J. Ophthalmol.	1260	5.258	Q1	3	Am. J. Ophthalmol.	67,517	5.258	Q1
4	Graef Arch. Clin. Exp.	1033	3.117	Q2	4	Retina-J. Ret. Vit. Dis.	48,182	4.256	Q1
5	Brit. J. Ophthalmol.	948	4.638	Q1	5	Acta Ophthalmol.	38,288	3.761	Q1
6	Ophthalmology	800	12.079	Q1	6	Brit. J. Ophthalmol.	35,172	4.638	Q1
7	Eur. J. Ophthalmol.	666	2.597	Q3	7	Graef Arch. Clin. Exp.	19,714	3.117	Q2
8	Eye	651	3.775	Q1	8	Eye	14,662	3.775	Q1
9	PLoS ONE	585	3.240	Q2	9	PLoS ONE	11,479	3.240	Q2
10	Acta Ophthalmol.	535	3.761	Q1	10	Prog. Retin. Eye Res.	10,672	21.198	Q1

## Data Availability

Not applicable.

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
