# Peer review of "Research Trends and Hotspots of Retinal Optical Coherence Tomography: A 31-Year Bibliometric Analysis"

_jcm, 2022, doi:10.3390/jcm11195604_

Round 1

Reviewer 1 Report

In this review article, the authors summarized 31 years of retinal OCT research, shedding light on the hotspots, main themes and emerging frontiers to guide future research.

This paper is well organized. On the other hand, the results presented are consistent with clinical impressions, and there may be few new findings for the reader. For example, Discussion 4.2.1, Cluster 1. Thickness Measurements by OCT (Page 10, line 245).

This is not a surprising result because the advent of OCT has made it possible to accurately quantify retinal thickness. Therefore, the authors conclude that the present results will guide the direction of future research, but not necessarily so.

There are no other points raised in the text other than the one below.

Discussion 4.2.2, Cluster 2 (Page 11, line 268). There are other diseases besides DME that can cause ME. It would be desirable to add a discussion of those as well. 

Reviewer 2 Report

Optical Coherence Tomography (OCT) is a powerful technique for analyzing the different layers of the retina in vivo and is routinely used in clinics as well as in retina research. Hence, this bibliometric analysis by Lin et al. can be interesting for the community, especially since it covers publications for 31 years. The list of keywords and the combinations thereof used to search for the data is comprehensive. The manuscript is well written; the figures are clearly presented and easy to follow. 

1. I have only one suggestion, that can potentially make this study more interesting from the reader’s perspective, as some of the conclusions are obvious, especially when 30 years of data is analyzed and represented as a whole. Example Figure 2. Co-occurrence of keywords – This figure shows the co-occurrence of keywords as 5 clusters in 31 years. It would be interesting if the authors made a separate analysis for publications for each decade – Figure 2A – 1991-2001, Figure 2B – 2002-2011, Figure 2C – 2012-2021. Comparison of these three networks will help answer following questions –

-  How are the hotspots and trends changing over time?

-  Are certain co-occurrences changing or staying the same? Why?

- Is OCT getting more automated – the bubbles for keywords like ‘high-speed’, ‘segmentation’ may get larger and others like computation, automation, artificial intelligence may appear more prominently. 

2. It is not clear if the studies included are from clinical studies (involving humans) or from research-based studies (involving animals). Were the authors able to make a distinction? This needs to be clarified. 
